# A simpler screening tool for sarcopenia in surgical patients

Onuma Chaiwat[1,2]*, Mingkwan Wongyingsinn[1], Weerasak Muangpaisan[2,3], Chalobol Chalermsri[3], Arunotai Siriussawakul[1,2], Pornpoj Pramyothin[4], Poungkaew Thitisakulchai[5], Panita Limpawattana[6], Chayanan Thanakiattiwibun[2]

**1** Department of Anesthesiology, Faculty of Medicine Siriraj Hospital, Mahidol University, Bangkok, Thailand, **2** Integrated Perioperative Geriatric Excellent Research Center, Faculty of Medicine Siriraj Hospital, Mahidol University, Bangkok, Thailand, **3** Department of Preventive and Social Medicine, Faculty of Medicine Siriraj Hospital, Mahidol University, Bangkok, Thailand, **4** Division of Nutrition, Department of Medicine, Faculty of Medicine Siriraj Hospital, Mahidol University, Bangkok, Thailand, **5** Department of Rehabilitation Medicine, Faculty of Medicine Siriraj Hospital, Mahidol University, Bangkok, Thailand, **6** Division of Geriatric Medicine, Department of Internal Medicine, Faculty of Medicine, Khon Kaen University, Khon Kaen, Thailand

* onuma.cha@mahidol.ac.th

**Data Availability Statement:** All relevant data are within the paper and its Supporting Information files.

## Abstract

### Background

Sarcopenia is defined as decreased skeletal muscle mass and muscle functions (strength and physical performance). Muscle mass is measured by specific methods, such as bioelectrical impedance analysis and dual-energy X-ray absorptiometry. However, the devices used for these methods are costly and are usually not portable. A simple tool to screen for sarcopenia without measuring muscle mass might be practical, especially in developing countries. The aim of this study was to design a simple screening tool and to validate its performance in screening for sarcopenia in older adult cancer patients scheduled for elective surgery.

### Methods

Cancer surgical patients aged >60 years were enrolled. Their nutritional statuses were evaluated using the Mini Nutrition Assessment-Short Form. Sarcopenia was assessed using Asian Working Group for Sarcopenia (AWGS) criteria. Appendicular skeletal muscle mass was measured by bioelectrical impedance analysis. Four screening formulas with differing combinations of factors (muscle strength, physical performance, and nutritional status) were assessed. The validities of the formulas, compared with the AWGS definition, are presented as sensitivity, specificity, accuracy, and area under a receiver operating characteristic curve.

### Results

Of 251 enrolled surgical patients, 84 (34%) were diagnosed with sarcopenia. Malnutrition (odds ratio [OR]: 2.89, 95% CI: 1.40–5.93); underweight status (OR: 2.80, 95% CI: 1.06–7.43); and age increments of 5 years (OR: 1.78, 95% CI: 1.41–2.24) were independent predictors of preoperative sarcopenia. The combination of low muscle strength and/or

**Funding:** This research project was supported by Faculty of Medicine Siriraj Hospital, Mahidol University, Grant Number (IO) R016034004. The funders had no role in study design, data collection, and analysis, decision to publish, or preparation of the manuscript.

**Competing interests:** The authors have declared that no competing interests exist.

abnormal physical performance, plus malnutrition/risk of malnutrition had the highest sensitivity, specificity, and accuracy (81.0%, 78.4%, and 79.3%, respectively). This screening formula estimated the probability of sarcopenia with a positive predictive value of 65.4% and a negative predictive value of 89.1%.

## Conclusion

Sarcopenia screening can be performed using a simple tool. The combination of low muscle strength and/or abnormal physical performance, plus malnutrition/risk of malnutrition, has the highest screening performance.

## Background

According to the United States Census Bureau, 20% of Americans are predicted to be aged greater than 65 years in 2030, and 50% of them will require an operation [1]. In the case of Thailand, it is projected that 26.6% of the population will be aged over 60 years in 2030 [2]. Aging is associated with an increasing prevalence of frailty, comorbidities, a decline in functional reserve, and sarcopenia. Sarcopenia has been repeatedly demonstrated to be one of the strongest predictors of both short- and long-term outcomes following complicated surgical procedures [3]. Even though surgery is the most effective cancer therapy, complication rates and mortality increase among older adult patients, and this can lessen the advantage of oncological therapy [4]. Different definitions of sarcopenia have been utilized by research groups around the world [5–11], such as the European Working Group on Sarcopenia in Older People (EWGSOP) in 2010, the Asian Working Group for Sarcopenia (AWGS) in 2014, and the Japan Society of Hepatology (JSH) in 2016. In essence, each definition proposed to date defines sarcopenia as a state of decreased skeletal muscle mass and muscle function. Muscle function can be divided into those that require both muscle strength and physical performance, or only one of those elements [12]. However, skeletal muscle mass is mainly used as the core element of all definitions. In early 2018, the EWGSOP met again (EWGSOP2) to revise the definition and diagnosis of sarcopenia. The updated EWGSOP2 consensus targeted low muscle strength as the first key component of sarcopenia, confirmed sarcopenia diagnoses by low muscle quantity and/or quality, and identified poor physical performance as indicative of severe sarcopenia [13]. The recently updated 2019 AWGS consensus contains revisions to the diagnostic algorithm, the protocols, and some criteria, including the cutoff values for low muscle strength and low physical performance. Nevertheless, skeletal muscle strength and mass remain foundational to a definitive clinical diagnosis of sarcopenia [14].

As regards the assessment of muscle mass for a diagnosis of sarcopenia, muscle mass is commonly assessed by bioelectrical impedance analysis (BIA) or dual-energy X-ray absorptiometry (DXA) [15]. BIA is a practical and portable method that does not expose a patient to any radiological harm [16]. Although the use of analyzers and absorptiometers were the main methods previously recommended for the assessment of body composition [17], both devices have some limitations in terms of their accessibility and cost [18]. A recent study demonstrated that the use of anthropometric data, such as body mass index (BMI), was an indirect means of measuring body composition that produced results comparable with those obtained with DXA [17]. Malnutrition in hospitalized patients, especially cancer patients, was documented with a prevalence up to 50% [19]. Malnourished surgical patients were reported to have a postoperative morbidity rate as high as 33%, with outcomes that included poor wound healing, increased

postoperative infection, overgrowth of bacteria in the gastrointestinal tract, delayed return of recovery function, and prolonged hospital stay [20–24]. Since malnutrition and malignancy are factors contributing to the development of sarcopenia [13], a simple tool to screen for sarcopenia in patients who have cancer might be possible by including malnutrition and an underweight status as screening factors.

In addition, several screening tools for sarcopenia have been introduced. The EWGSOP2 recommends the use of the SARC-F questionnaire to elicit self-reported signs and symptoms that are characteristic of sarcopenia [13]. Calf circumference was incorporated in SARc-CalF as an additional parameter to enable indirect measuring of muscle mass [25, 26]. The Ishii model was also developed to estimate the probability of sarcopenia in older community-dwelling adults [27]. However, no study has reported the superiority of any tool over the others because no head-to-head comparison study has been performed.

The aims of this study were to design a simple screening tool and to validate its performance in screening for sarcopenia in older adult cancer patients prior to undergoing elective surgery.

## Materials and methods

### Design

This prospective longitudinal study was conducted at Siriraj Pre-anesthesia Assessment Center (SiPAC), Department of Anesthesiology, Faculty of Medicine Siriraj Hospital, Mahidol University between April 2017 and December 2017. Siriraj Hospital is a 2300-bed, university-based, national tertiary-referral hospital in Bangkok, Thailand. All patients or their legal guardians provided informed consent in writing. The Siriraj Institutional Review Board approved the study protocol (SIRB COA no. Si 101/2017). The study was registered with the Thai Clinical Trials Registry (TCTR20181223002).

### Study population

The study population comprised cancer patients aged older than 60 years who presented at SiPAC prior to undergoing elective surgery. Individuals unable to walk or stand unaided were excluded because the BIA device could measure muscle mass only while patients were in the standing position. Patients were also excluded if they had one or more of the following: limitations revealed by BIA; a pacemaker; the use of a medication, herb, or hormones that affect muscle mass and strength (eg, estrogen, testosterone, eltroxin, and steroid); and alcohol consumption or strenuous exercise during the 12 hours preceding the scheduled BIA. Patients meeting the selection criteria were invited to participate. After providing written informed consent, the enrolled patients underwent preoperative nutritional screening and sarcopenia assessment.

### Measurement instruments and data collection

**Preoperative patient characteristic data** were collected. Details of the following were recorded: gender, age, body weight, American Society of Anesthesiologists (ASA) physical status, underlying medical problems, current medications, smoking status, alcohol consumption status, surgical services, diagnosis, operation, and preoperative preparation.

**Preoperative nutritional screening** was performed using the Mini Nutritional Assessment–Short Form (MNA-SF). The items examined were reduction in dietary intake and weight loss during the preceding 3 months, BMI, mobility, psychological stress or acute disease during the preceding 3 months, and neuropsychological problems [28, 29]. The maximum

score was 14 points. Nutritional status was reported as "normal" (12–14 points), "at risk of malnutrition" (8–11 points), and "malnutrition" (0–7 points).

**Assessment of sarcopenia** was performed by measuring (1) the appendicular skeletal muscle mass, with a BIA device (Tanita MC-780U Multi Frequency Segmental Body Composition Analyzer; Tanita Corporation, Tokyo, Japan); (2) muscle grip strength of the dominant hand at maximum strength, using a handgrip dynamometer (TKK 5401 Grip D; Takei Scientific Instruments Co., Ltd., Niigata, Japan); and (3) physical performance, using the 6-meter walk test [5, 30]. For the walk test, participants stood with their feet behind a starting line, and started walking while following the examiner's instructions. Timing started with the first step and stopped when the patient's first foot completely crossed the 6-meter line. The AWGS recommends the following cutoff values for sarcopenia diagnoses: low handgrip strength: <26 kg for men, and <18 kg for women; and low physical performance: gait speed <0.8 m/s [9]. For muscle mass measurement in the Thai population, the Thai National Guideline for the Management of Geriatric Syndromes, Frailty and Sarcopenia defines low muscle mass as <7.9 kg/m$^2$ in men and <6.0 kg/m$^2$ in women [31]. A flowchart of the malnutrition and sarcopenia screening is presented in Fig 1.

A diagnosis of sarcopenia was based on documented **low muscle mass** plus **low muscle strength** or **low physical performance** [30]. Patients with low muscle mass, low muscle strength, and low physical performance were classed as having severe sarcopenia [32].

**Outcome measures** were scores for the Barthel Index of Activities of Daily Living at 3 months and 1 year after sarcopenia screening; any infections in the hospital; length of hospital stay; hospital mortality; and mortality at 3 months and 1 year after sarcopenia screening.

## Design of a simple tool for sarcopenia diagnosis

Four formulas for the diagnosis of sarcopenia were developed. They used differing combinations (C1, C2, C3, and C4) of factors deemed to be relevant to the diagnosis of sarcopenia related to cancer (Table 1). The factors were "muscle strength", "physical performance", "risk of malnutrition", "malnutrition", and "underweight BMI". Muscle mass was not used as a factor. Formula 1 used the combination of low muscle strength and abnormal physical performance (C1). Formula 2 used low muscle strength; abnormal physical performance; and malnutrition/risk of malnutrition (C2). Formula 3 used low muscle strength and/or abnormal physical performance, plus malnutrition/risk of malnutrition (C3). Formula 4 used low muscle strength and/or abnormal physical performance, plus underweight BMI (C4). The 2014 AWGS criteria [9] and the updated 2019 AWGS [14] criteria were used as gold standards for the sarcopenia diagnoses. The EWGSOP2 criteria were then compared to both AWGS versions.

## Statistical analysis

The sample size estimate was based on a reported 30% prevalence of sarcopenia among community-dwelling, older adult Thais [33]. Using a 6% error, a minimum sample size of 225 cases was calculated. To compensate for a possible 10% dropout rate, the size was increased to 250.

Demographic and clinical variables are summarized using descriptive statistics. Continuous variables are described as mean and standard deviation (SD) or median and interquartile range (IQR), depending on the data distribution. Normality was checked using a histogram and the Kolmogorov–Smirnov test. Categorical variables are described as frequency and percentage. The prevalences of sarcopenia and malnutrition are presented as percentage. Comparisons between the sarcopenia and non-sarcopenia groups were performed using the

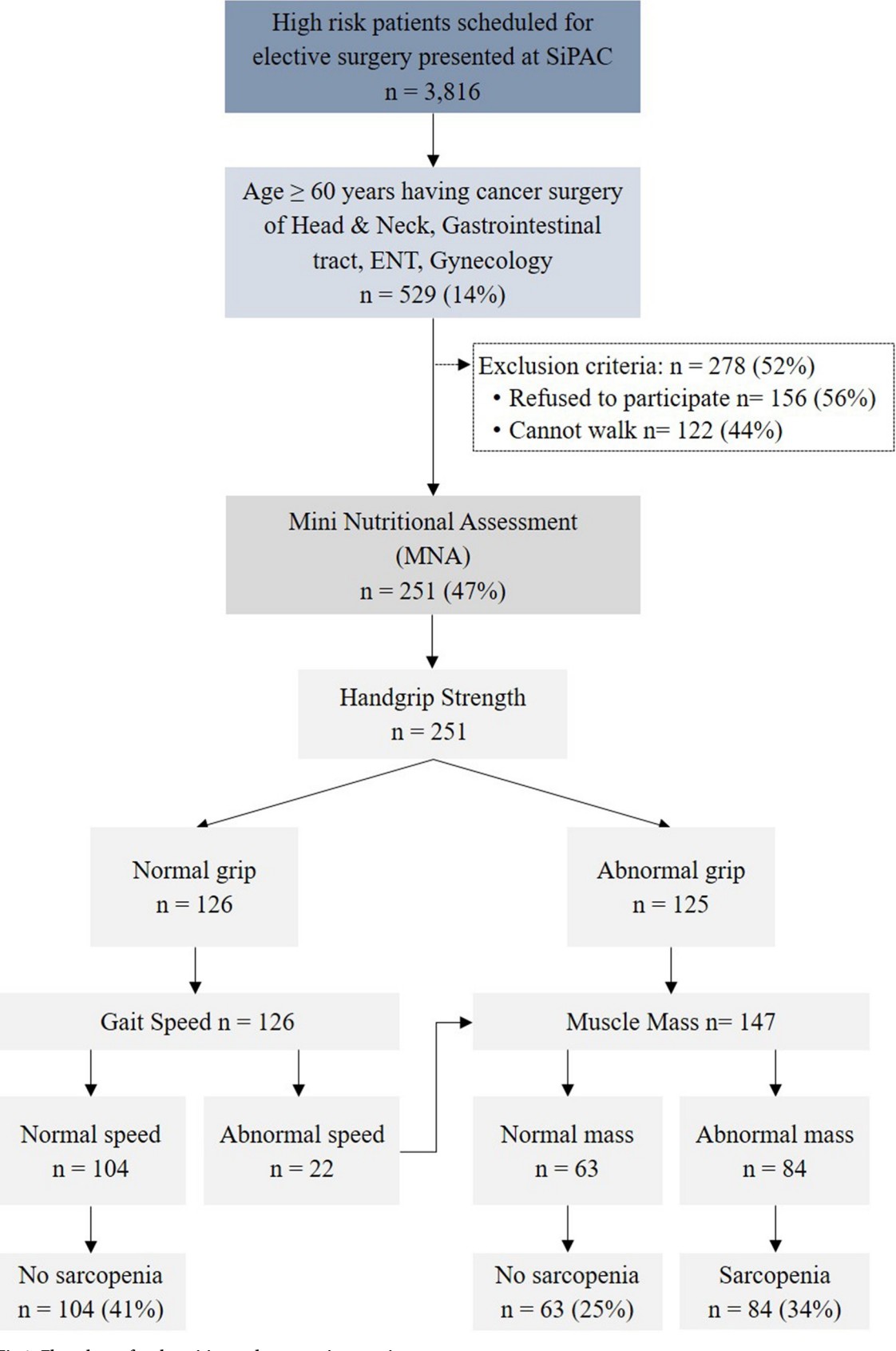

**Fig 1. Flow chart of malnutrition and sarcopenia screening.**

**Table 1. Combinations used to diagnose sarcopenia.**

| | Muscle mass | Muscle strength | | Physical performance | | Malnutrition/ Risk of malnutrition | | BMI |
|---|---|---|---|---|---|---|---|---|
| AWGS | ↓ | ↓ | and/or | ↓ | | – | | – |
| EWGSOP2 | ↓ | ↓ | | – | | – | | – |
| C1 | – | ↓ | and | ↓ | | – | | – |
| C2 | – | ↓ | and | ↓ | and | ↓ | | – |
| C3 | – | ↓ | and/or | ↓ | and | ↓ | | – |
| C4 | – | ↓ | and/or | ↓ | | – | and | ↓ |

Abbreviations: AWGS, Asian Working Group for Sarcopenia (low muscle mass, and low muscle strength and/or low physical performance); BMI, body mass index; C1, combination of muscle strength and physical performance; C2, combination of muscle strength, physical performance, and malnutrition/risk of malnutrition; C3, combination of muscle strength and/or physical performance, plus malnutrition/risk of malnutrition; C4, combination of muscle strength and/or physical performance, plus body mass index; EWGSOP2, European Working Group on Sarcopenia in Older People 2 (low muscle mass and low muscle strength).

independent $t$-test or Mann–Whitney U test for continuous variables, and the chi-squared test or Fisher's exact test for categorical variables. Factors associated with sarcopenia were identified using logistic regression. The risk factors with a univariable $P$ value less than 0.2 were entered into multiple logistic regression. They were gender, age, BMI, ASA status, diabetes mellitus (DM), hypertension, dyslipidemia (DLP), chronic kidney disease/end-stage renal disease (CKD/ESRD), current smoker, alcohol consumption, preoperative Barthel Index score <70, MNA-SF, waiting time for surgery, and infection. With those factors, a multivariate logistic regression analysis with enter elimination was utilized to appraise the independent variables associated with preoperative sarcopenia. The validities of the 4 formulas for diagnosis of sarcopenia were assessed relative to the AWGS definition. The diagnostic performances were evaluated in terms of sensitivity, specificity, positive predictive value (PPV), negative predictive value (NPV), positive likelihood ratio (LR+), negative likelihood ratio (LR-), accuracy, and area under a receiver operating characteristic curve (AUROC). Statistical analyses were performed using PASW Statistics for Windows (version 18.0; SPSS Inc., Chicago, IL, USA) (S1 File).

## Results

### Diagnosis of sarcopenia

In all, 3816 patients presented at SiPAC between April 2017 and December 2017, with 840 (22.0%) being diagnosed with cancer. Of that cancer group, 529 patients (62.9%) were aged greater than 60 years. Two hundred and seventy-eight (278) patients were excluded because they declined to participate (56%) or were unable to walk (44%). The remaining 251 surgical patients were enrolled. Eighty-four (34%) were diagnosed with sarcopenia as per the AWGS criteria. Fig 1 presents a flowchart of the malnutrition and sarcopenia screening process. Eighty-four subjects (34%) had an abnormal walking speed, and another 63 (25%) had an abnormal grip strength. Those 147 patients were then subjected to BIA. Of those, only 84 demonstrated a muscle mass below the recommended cutoff. This gave a sarcopenia prevalence of 34%. Based on the EWGSOP conceptual stages of sarcopenia [5], 40% (34/84 patients) of our sarcopenic patients were categorized as having severe sarcopenia. The prevalence of sarcopenia increased with advancing age, reaching 70% in the patients aged over 80 years. Eleven patients (13%) with preoperative sarcopenia experienced a change in their treatment plan; 82% (9/11 patients) were receiving palliative care; and 18% (2/11 patients) were receiving radiation therapy. In the non-sarcopenia group, only 5 patients (3%) experienced a change in their treatment

plan; 80% (4/5 patients) were undergoing chemotherapy; and 20% (1/5 patients) was receiving palliative care. The remaining 235 patients were included in the outcome measurement analysis.

Our analysis revealed that 123 patients (49%) were at risk for malnutrition, while 28 (11%) had malnutrition. Among the 84 sarcopenic patients, 60% were at risk for malnutrition, and 21% had malnutrition.

### Factors associated with sarcopenia

The demographic, anthropometric, clinical, and surgical characteristics of the study patients are detailed in Table 2. Relative to the non-sarcopenic patients, the sarcopenic patients were significantly older, had a higher proportion who were underweight, and a lower proportion who were overweight or obese. There was no statistically significant difference in the Charlson Comorbidity Index of the groups; however, the sarcopenic group had significantly fewer patients with DM and DLP. A significantly higher percentage of patients with sarcopenia demonstrated a moderate to severe disability or malnutrition before surgery (Table 2). There was no significant difference between the groups in terms of choice of anesthesia, duration of anesthesia, or total blood loss. The multivariate analysis (Table 3) revealed that the independent predictors of preoperative sarcopenia were malnutrition (odds ratio [OR]: 2.89, 95% confidence interval [CI]: 1.40–5.93); an underweight status (OR: 2.80, 95% CI: 1.06–7.43); and age increments of 5 years (OR: 1.78, 95% CI: 1.41–2.24). Being overweight was found to be a protective factor against preoperative sarcopenia (OR: 0.19, 95% CI: 0.08–0.47).

### Screening performance of the tool

Formula-combination C3 (low muscle strength and/or abnormal physical performance, plus malnutrition/risk of malnutrition) demonstrated the highest sensitivity and accuracy when using the 2014 AWGS or the updated 2019 AWGS criteria as the gold standards. The sensitivity, specificity, accuracy, and AUROC of formula-combination C3 were 81.0%, 78.4%, 79.3%, and 0.8, respectively when using the 2014 AWGS criteria as the gold standard. The C3 formula presented the ability to estimate the probability of sarcopenia, with a PPV of 65.4% and an NPV of 89.1%. EWGSOP2 demonstrated the highest specificity (100%) for diagnosis of sarcopenia (Table 4A). The prevalence of sarcopenia by EWGSOP2 definition was 28%. In addition, EWGSOP2 proposed the term "probable sarcopenia", for which the diagnosis requires only lower muscle strength. The prevalence of probable sarcopenia was 49.8%; however, muscle mass measurement was needed to confirm a diagnosis of sarcopenia. Regarding the screening performance of formula-combination C3 in males, that formula showed a high sensitivity, specificity, PPV, and NPV (72.7%, 91.1%, 83.3%, and 86.5%, respectively). As to its screening performance in females, while it demonstrated a high sensitivity (96.6%) and high NPV (98%), formula-combination C3 had a low specificity (63.3%) and low PPV (50.0%). The sensitivity, specificity, accuracy, and AUROC of formula C3 was 80%, 68.5%, 73.3%, and 0.74, respectively, when using the updated 2019 AWGS criteria as the gold standard (Table 4B).

### Sarcopenia and clinical outcomes

Three months after hospital discharge, the sarcopenic patients demonstrated a higher incidence of moderate to severe disability than the non-sarcopenic patients (22% vs. 8%, $P = 0.006$). However, there was no significant difference 1 year after the sarcopenia screening (sarcopenic and non-sarcopenic surgical patients: 8.8% vs. 2.9%, $P = 0.127$). As to the mortality rates, there was no significant difference in the rates 3 months after the sarcopenia screening.

**Table 2. Demographic, clinical, and surgical characteristics and outcomes.**

| Characteristics | All patients | Non-sarcopenic | Sarcopenic | *P* value |
|---|---|---|---|---|
| | (n = 251) | (n = 167) | (n = 84) | |
| Male gender | 145 (57.8%) | 90 (53.9%) | 55 (65.5%) | 0.104 |
| Age (years) | 71.6±7.6 | 69.6±6.3 | 75.5±8.3 | <0.001 |
| BMI (kg/m$^2$) | 23.4±4.4 | 24.8±4.3 | 20.6±2.9 | <0.001 |
| BMI category: | | | | <0.001 |
| Underweight (<18.5) | 29 (11.6%) | 9 (5.4%) | 20 (23.8%) | |
| Normal weight (18.5–24.9) | 138 (54.9%) | 82 (49.1%) | 56 (66.7%) | |
| Overweight (25.0–29.9) | 69 (27.5%) | 61 (36.5%) | 8 (9.5%) | |
| Obesity (≥30.0) | 15 (5.9%) | 15 (9.0%) | 0 (0.0%) | |
| ASA classification: | | | | 0.074 |
| ≤2 | 157 (62.5%) | 111 (66.5%) | 46 (54.8%) | |
| >2 | 94 (37.5%) | 56 (33.5%) | 38 (45.2%) | |
| Underlying medical problem: | | | | |
| DM | 75 (29.9%) | 59 (35.3%) | 16 (19.0%) | 0.008 |
| HT | 157 (62.5%) | 112 (67.1%) | 45 (53.6%) | 0.039 |
| DLP | 129 (51.4%) | 95 (56.9%) | 34 (40.5%) | 0.016 |
| CVA | 11 (4.4%) | 6 (3.6%) | 5 (6.0%) | 0.518 |
| CKD/ESRD | 25 (9.9%) | 14 (8.4%) | 11 (13.1%) | 0.170 |
| Charlson Comorbidity Index | 4.1±2.3 | 4.1±2.3 | 4.0±2.3 | 0.752 |
| Chemotherapy | 8 (3.2%) | 7 (4.2%) | 1 (1.2%) | 0.274 |
| Radiotherapy | 2 (0.8%) | 0 | 2 (2.4%) | 0.111 |
| Current smoker | 120 (47.8%) | 73 (43.7%) | 47 (56.0%) | 0.082 |
| Alcohol consumption: | | | | 0.183 |
| None | 173 (68.9%) | 118 (70.7) | 55 (65.5) | |
| Habitual | 53 (21.1%) | 30 (18.0%) | 23 (27.4%) | |
| Social | 25 (9.9%) | 19 (11.4%) | 6 (7.1%) | |
| **Preoperative data** | | | | |
| Surgical service: | | | | 0.643 |
| GI | 68 (27.1%) | 44 (26.3%) | 24 (28.6%) | |
| URO | 70 (27.9%) | 45 (26.9%) | 25 (29.8%) | |
| GYN | 25 (9.9%) | 20 (12.0%) | 5 (6.0%) | |
| HNB | 34 (13.5%) | 24 (14.4%) | 10 (11.9%) | |
| ENT | 40 (15.9%) | 24 (14.4%) | 16 (19.0%) | |
| Other | 14 (5.6%) | 10 (6.0%) | 4 (4.8%) | |
| Barthel index score ≤70 | 6 (2.4%) | 1 (0.6%) | 5 (6.2%) | 0.017 |
| Malnutrition | 150 (59.8%) | 82 (49.1%) | 68 (81.0%) | <0.001 |
| Waiting time for surgery (days) | 24 (13–39) | 25 (15–40) | 22 (10–38) | 0.129 |
| Severity of cancer: | | | | |
| Distant organ metastasis | 50 (19.9) | 30 (18.0) | 20 (23.8) | 0.316 |
| **Intraoperative data** | | | | |
| Duration of anesthesia (min) | 227.9±164.1 | 229.1±158.6 | 225.2±176.9 | 0.868 |
| Blood loss (ml) | 65 (15–300) | 70 (20–300) | 50 (10–300) | 0.470 |
| Electrolyte imbalance | 23 (9.8%) | 15 (9.3%) | 8 (11.0%) | 0.644 |
| Infection: | | | | 0.193 |
| None | 206 (87.7%) | 145 (89.5%) | 61 (83.6%) | |
| Sepsis | 4 (1.7%) | 2 (1.2%) | 2 (2.7%) | |
| Wound | 3 (1.3%) | 1 (0.6%) | 2 (2.7%) | |

(*Continued*)

**Table 2.** (Continued)

| Characteristics | All patients | Non-sarcopenic | Sarcopenic | P value |
|---|---|---|---|---|
| | (n = 251) | (n = 167) | (n = 84) | |
| Respiratory tract | 7 (3.0%) | 3 (1.9%) | 4 (5.5%) | |
| Urinary tract | 4 (1.7%) | 2 (1.2%) | 2 (2.7%) | |
| Others | 11 (4.7%) | 9 (5.6%) | 2 (2.7%) | |
| **Outcomes** | | | | |
| Length of stay (days) | 6 (4–9) | 6 (4–9) | 6 (4.5–9.5) | 0.198 |
| Hospital mortality rate | 3 (1.3%) | 2 (1.2%) | 1 (1.3%) | 1.000 |
| Barthel Index score ≤70 at 3 months after hospital discharge (n = 239) | 30 (12.6%) | 13 (8.0%) | 17 (22.1%) | 0.006 |
| Mortality rate at 3 months after screening (n = 251) | 12 (4.8%) | 5 (3.0%) | 7 (8.3%) | 0.061 |
| Barthel Index score ≤70 at 1 year after screening (n = 194) | 9 (4.6%) | 4 (2.9%) | 5 (8.8%) | 0.127 |
| Mortality rate at 1 year after screening (n = 250) | 56 (22.4%) | 29 (17.5%) | 27 (32.1%) | 0.010 |

Data are presented as number and percentage, mean ± standard deviation, or median and interquartile range.

A P value<0.05 indicates statistical significance.

Malnutrition was assessed by MNA-SF (Mini Nutritional Assessment–Short Form).

Abbreviations: ASA, American Society of Anesthesiologists; BMI, body mass index; CAD, coronary artery disease; CKD, chronic kidney disease; COPD, chronic obstructive pulmonary disease; CVA, cerebrovascular accident; DLP, dyslipidemia; DM, diabetes mellitus; ENT, ear, nose, and throat; ESRD, end-stage renal disease; GA, general anesthesia; GI, gastrointestinal; GYN, gynecology; HNB, head, neck, and breast; HT, hypertension; RA, regional anesthesia; URO, urology.

In contrast, the sarcopenic patients had a significantly higher mortality than the non-sarcopenic patients 1 year after the screening (32% vs. 17.5%, *P* = 0.01; Table 2).

## Discussion

Although advances in perioperative management and surgical techniques for oncology patients have reduced postoperative complications, challenges remain for older adults. This is due to the declines in their functional reserves, worsening frailty, and increasing incidence of

**Table 3. Independent risk factors associated with preoperative sarcopenia.**

| Factors | Adjusted odds ratio (95% CI) | P value |
|---|---|---|
| Age (years, 5 units) | 1.78 (1.41–2.24) | <0.001 |
| Body mass index: | | |
| Normal | 1 | |
| Underweight | 2.80 (1.06–7.43) | 0.038 |
| Overweight | 0.19 (0.08–0.47) | <0.001 |
| Malnutrition | 2.89 (1.40–5.93) | 0.004 |
| Pre-Barthel Index score ≤70 | 10.48 (0.84–122.08) | 0.061 |

Adjusted for gender, American Society of Anesthesiologists (ASA) Classification, diabetes mellitus, hypertension, dyslipidemia, chronic kidney disease/end-stage renal disease, current smoker, alcohol consumption, waiting time for surgery, and infection.

A P value <0.05 indicates statistical significance.

Malnutrition was assessed by MNA⁻SF (Mini Nutritional Assessment–Short Form).

Abbreviation: CI, confidence interval.

**Table 4. Validity of combinations used to diagnose sarcopenia.**

**a.** Validity based on 2014 AWGS criteria as gold standard

| Tools | Sensitivity (95% CI) | Specificity (95% CI) | PPV (95% CI) | NPV (95% CI) | LR+ (95% CI) | LR- (95% CI) | Accuracy (95% CI) | AUROC (95% CI) |
|---|---|---|---|---|---|---|---|---|
| EWGSOP2 | 84.5% (74.9–91.5) | 100.0% (97.8–100) | – | 92.8% (88.6–95.5) | – | 0.15 (0.09–0.26) | 94.8% (91.3–97.2) | 0.70 (0.63–0.78) |
| C 1 | 40.5% (29.9–51.8) | 83.2% (76.7–88.6) | 54.8% (44.2–65.0) | 73.5% (69.7–77.1) | 2.4 (1.6–3.7) | 0.7 (0.6–0.9) | 68.9% (62.8–74.6) | 0.62 (0.54–0.70) |
| C 2 | 35.7% (25.6–46.9) | 90.4% (84.9–94.4) | 65.2% (52.0–76.4) | 73.7% (70.3–76.8) | 3.7 (2.2–6.4) | 0.7 (0.6–0.8) | 72.1% (66.1–77.6) | 0.63 (0.55–0.71) |
| C 3 | **81.0%** (70.9–88.7) | **78.4%** (71.4–84.4) | **65.4%** (58.1–72.0) | **89.1%** (84.0–92.8) | **3.8** (2.8–5.1) | **0.2** (0.2–0.4) | **79.3%** (73.7–84.1) | **0.80** (0.74–0.86) |
| C 4 | 23.8% (15.2–34.4) | 98.8% (95.7–99.9) | 90.9% (70.5–97.7) | 72.1% (69.6–74.4) | 19.9 (4.8–83.1) | 0.8 (0.7–0.9) | 73.7% (67.8–79.0) | 0.61 (0.54–0.69) |

**b.** Validity based on 2019 AWGS criteria as gold standard

| Tools | Sensitivity (95% CI) | Specificity (95% CI) | PPV (95% CI) | NPV (95% CI) | LR+ (95% CI) | LR- (95% CI) | Accuracy (95% CI) | AUROC (95% CI) |
|---|---|---|---|---|---|---|---|---|
| EWGSOP2 | 76.2% (66.9–84.0) | 100% (97.5–100.0) | – | 85.4% (80.6–89.2) | – | 0.24 (0.17–0.34) | 90.0% (85.7–93.5) | 0.88 (0.83–0.93) |
| C 1 | 59.1% (49.0–68.6) | 64.4% (56.0–72.1) | 54.4% (47.7–61.0) | 68.6% (62.8–73.9) | 1.66 (1.3–2.2) | 0.6 (0.5–0.8) | 62.2% (55.8–68.2) | 0.62 (0.55–0.69) |
| C 2 | 49.5% (39.6–59.5) | 79.5% (72.0–85.7) | 63.4% (54.4–71.6) | 68.6% (64.0–72.9) | 2.4 (1.7–3.5) | 0.6 (0.5–0.8) | 66.9% (60.7–72.7) | 0.65 (0.57–0.72) |
| C 3 | **80%** (71.1–87.2) | **68.5%** (60.3–75.9 | **64.6%** (58.5–70.3) | **82.6%** (76.2–87.6) | **2.5** (2.0–3.3) | **0.3** (0.2–0.4) | **73.3%** (67.4–78.7) | **0.74** (0.68–0.81) |
| C 4 | 23.8% (16.0–33.1) | 97.3% (93.1–99.3) | 86.2% (69.2–94.6) | 64.0% (61.4–66.5) | 8.7 (3.1–24.2) | 0.8 (0.7–0.9) | 66.5% (60.3–72.3) | 0.61 (0.53–0.68) |

Abbreviations: C1, combination of muscle strength and physical performance; C2, combination of muscle strength, physical performance, and malnutrition/risk of malnutrition; C3, combination of muscle strength and/or physical performance, plus malnutrition/risk of malnutrition; C4, combination of muscle strength and/or physical performance, plus body mass index; CI, confidence interval; EWGSOP2, European Working Group on Sarcopenia in Older People 2 (low muscle strength and low muscle mass); LR+, positive likelihood ratio; LR-, negative likelihood ratio; NPV, negative predictive value; PPV, positive predictive value.

comorbidities [34]. It is therefore essential to identify high-risk older adult patients during the preoperative period. Since the patients in this study were all Thai, we applied the AWGS sarcopenia diagnosis criteria to identify sarcopenia in older-adult cancer patients presenting at SiPAC prior to undergoing elective surgery. The prevalence of sarcopenia in this population was 34%. A higher age, an underweight status, and malnutrition were found to be significantly associated with sarcopenia. Furthermore, this study demonstrated that a simple tool can be used to screen for sarcopenia in older-adult, surgical cancer patients without measuring muscle mass. The combination of low muscle strength and/or abnormal physical performance, plus malnutrition/risk of malnutrition (formula-combination C3), demonstrated high sensitivity, specificity, and predictive power when validated against a consensus of the AWGS.

Regarding the factors related to sarcopenia, we found older age, malnutrition, and an underweight status were significantly associated with sarcopenia. This was partially consistent with the findings of other studies. Khongsri et al. [33] reported that older age, low BMI, and low quadriceps strength were predictive factors for sarcopenia in community-dwelling, older adult Thais. However, older-adult cancer patients may be different from the general older-adult population because of their increased inflammatory response. This response leads to

cachexia, which exacerbates sarcopenia. This may explain why an underweight status was associated with sarcopenia in this study. Our nutritional status assessment using the MNA-SF revealed that 80% of the sarcopenic patients had malnutrition. Moreover, and importantly, we found that malnutrition was a strong predictor of preoperative sarcopenia (OR: 2.89, 95% CI: 1.4–2.9). Our results strongly suggest that sarcopenia and malnutrition should be considered and assessed together in surgical oncology patients. In addition, we demonstrated that the sarcopenic group had a significantly lower number of patients with DM and DLP. Hypothetically, sarcopenia should be related to higher DM and dyslipidemia due to the loss of metabolically active muscle tissue. However, sarcopenia was reported to be associated with lower DM if muscle mass was adjusted by height[2], but it was associated with a higher DM if the muscle mass was adjusted by weight or BMI [35]. In the current work, the unit of measurement used to measure muscle mass was kg/m$^2$.

The precise definition of sarcopenia varies from one research group to another [5, 6, 8–12]. Nevertheless, all definitions recommend that sarcopenia should be defined by a low muscle mass. This is commonly assessed by DXA, BIA, magnetic resonance imaging, and computed tomography [15, 27]. However, these tools are expensive, difficult to access, and are usually not portable. BIA is not routinely available in clinics and hospitals in developing countries. Research has been conducted on sarcopenia screening tools that do not require muscle mass to be measured, such as the SARC-F questionnaire, Ishii model, Goodman model, and anthropometric predictive equation models [13]. The SARC-F questionnaire has 5 items that are based on the cardinal features or consequences of sarcopenia [36]. Woo J et al. validated that questionnaire against 3 consensus definitions of sarcopenia from Europe, Asia, and an international group in Hong Kong. The questionnaire had excellent specificity (94%–99%) and NPV, but poor sensitivity [37]. Although SARC-F can be readily used in community healthcare and other clinical settings, it might be of limited value in rural areas and community hospitals in developing countries. This is because many patients might not be capable of self-reporting due to their very low levels of formal education. Interestingly, the Ishii model can estimate the probability of sarcopenia using the parameters of age, handgrip strength, and calf circumference in community-dwelling older adults at high risk for sarcopenia. The Ishii model demonstrated high sensitivity, specificity, PPV, and NPV when compared with EWGSOP [38]. However, the model was specifically developed for community-based older-adult Japanese, and it has not undergone external validation. It might therefore not be applicable to our particular population. The Goodman model is based on age and BMI, and it is employed as a screening tool to identify individuals likely to have low muscle mass and to benefit from a DXA scan. However, this screening tool was not specifically developed to screen for sarcopenia, whose current definitions include muscle strength measurement. It also has limitations when used with obese patients [27, 39]. Anthropometric predictive equation models assign scores based on routine clinical parameters such as weight, height, and gender [40, 41]. Although they can be used as a screening tool for sarcopenia in primary care settings, they have not yet been validated for hospital inpatients and non-Caucasian populations [27].

In view of the above constraints, our study developed a screening tool that did not involve the measurement of muscle mass. It was validated against the AWGS criteria, and its screening performance showed high sensitivity, specificity, PPV, and NPV. Its specificity and PPV values were slightly lower for females than males. Despite that, the tool represents a practical algorithm for the diagnosis of sarcopenia in older-adult, surgical cancer patients, and it does not need muscle mass to be measured (Fig 2). The algorithm starts with a case-finding phase. The C3 formula is applied, and surgical cancer patients with a high probability of having sarcopenia proceed to the next step. Step 2 involves screening. Muscle strength and physical performance (determined by walking speed) are measured, and nutritional status is evaluated with the

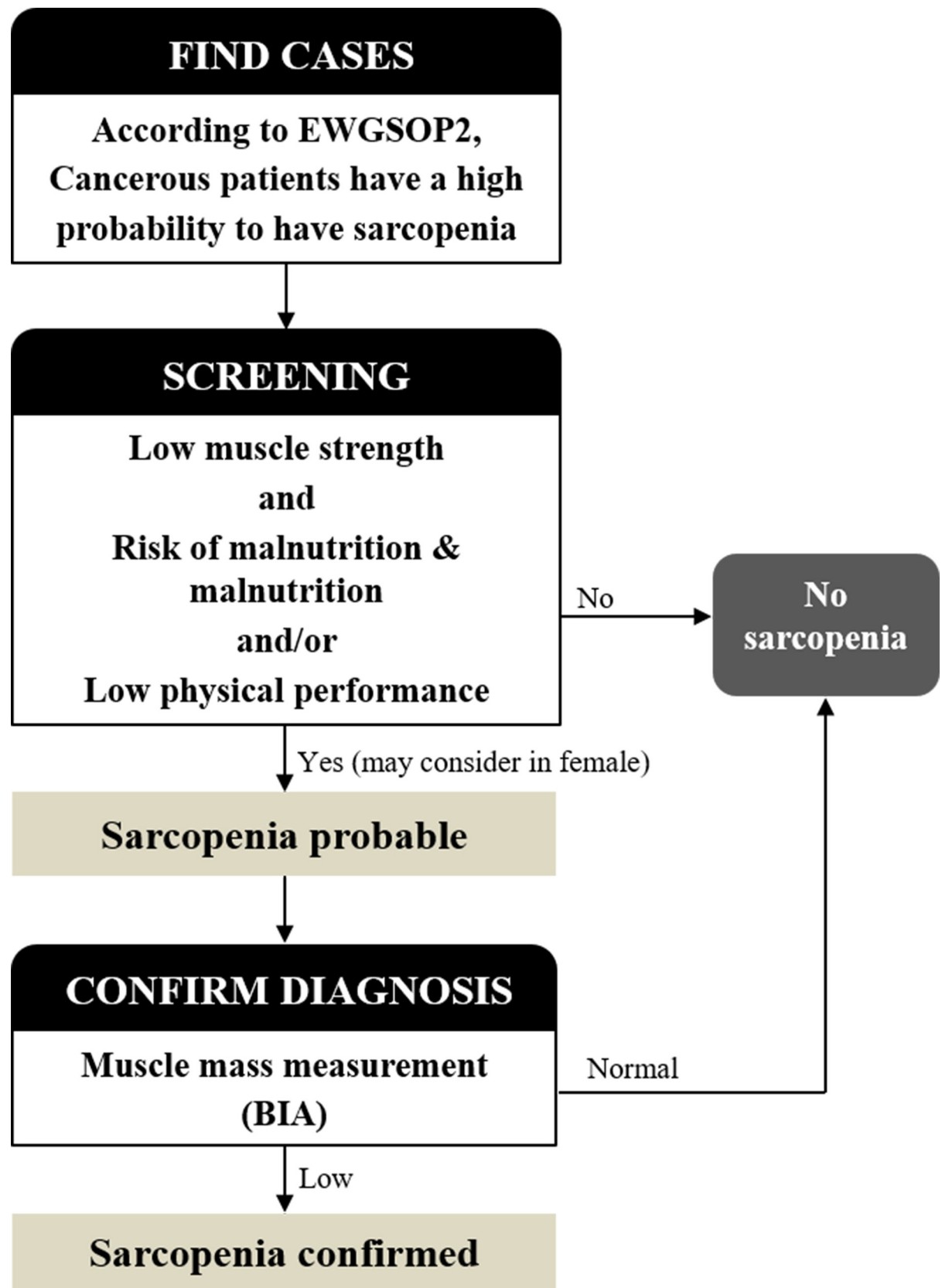

**Fig 2. Algorithm for proposed sarcopenia screening in older-adult, surgical oncology patients.**

MNA-SF. With male patients, sarcopenia is probable if they demonstrate a low muscle strength and/or low physical performance, and they are also assessed as being at risk of malnutrition or as having malnutrition. However, in the case of females meeting those conditions, sarcopenia should be considered as being only possible, given that the PPV for females is low (50%). As to the males and females who have negative findings in Step 2, they are deemed to not have sarcopenia. Step 3 is the confirmation phase. A BIA measurement is made of the muscle mass of the males and females who have positive findings in Step 2. If the mass is abnormal, sarcopenia is diagnosed. Conversely, if it is normal, sarcopenia is not diagnosed. The latter group of patients should be rescreened later.

With regard to the outcomes, Fukuda, *et al.* [34] found the incidence of severe postoperative complications (Clavien–Dindo grade >IIIa) to be significantly higher in a sarcopenic group than in a non-sarcopenic group (28.6% vs. 9.0%, $P = 0.03$) among gastrectomy patients [34]. In contrast, our study did not observe a significant difference in either the postoperative complications or the in-hospital mortality of the groups. Most patients in our cohort were not critically ill before undergoing surgery. Specifically, >60% of patients had an ASA score of <2, the average Charlson Comorbidity Index was only 4, and <3% received chemotherapy before surgery. We also found a greater decline in the activities of daily living in the sarcopenic patients 3 months after hospital discharge. It was suggested that an increased risk of physical limitation and disability in sarcopenic patients may adversely affect functional recovery, quality of life, and the independent performance of the activities of daily living [42].

This study has several limitations. Firstly, this was a single-center study that recruited patients from a preoperative assessment clinic. This suggests that our results may not be representative of, or generalizable to, all surgical patients at our center or in Thailand. Moreover, we used BIA to measure muscle mass, and only patients who could stand unaided were included. As a result, 21% of the patient candidates were excluded, which raises concerns about potential selection bias. Thirdly, the fat-free mass and body cell mass measured by BIA were calculated from the total body weight, using the assumption that 73% of the fat-free mass was water. Therefore, changes in the hydration state, such as edema, were the main limitation of this method [43]. In addition, we commenced this study before the publication of the updated AWGS recommendations on sarcopenia diagnoses. However, we used the revised diagnostic recommendations as another gold standard to validate our screening tool. The C3-combination of factors still demonstrated the highest sensitivity and accuracy. Furthermore, the cross-sectional design of our study meant that we were able to report that certain factors were found to be statistically significantly related to sarcopenia; however, we were not able to prove causation. Lastly, the proposed screening tool should be carefully interpreted with female patients because of its low specificity and PPV. In addition, as no external validation of the proposed screening tool was performed, its use with other populations is problematic. The strengths of this study are the prospective design, and the nutritional and sarcopenia assessments were performed by trained and experienced clinicians.

## Conclusions

A simple tool can be used to screen for sarcopenia in older-adult, surgical cancer patients without measuring muscle mass. The combination of low muscle strength and/or abnormal physical performance, plus malnutrition/risk of malnutrition demonstrated high sensitivity, specificity, PPV, and NPV. Preoperative screening of sarcopenia and malnutrition should be performed on all older-adult, surgical oncology patients to identify at-risk patients. This will enable prehabilitation and rehabilitation protocols covering nutritional and physical therapy to be implemented, thereby improving short- and long-term patient outcomes.

## Supporting information

**S1 File. Raw data of screening tool for sarcopenia.**
(XLSX)

## Acknowledgments

The authors gratefully acknowledge the patients who generously agreed to participate in this study; Asst. Prof. Dr. Chulaluk Komoltri for assistance with the statistical analyses; and Mrs. Dujprathana Pisalsarakij and Miss Tashita Pinsantia for assistance with the data collection. We also thank Mr. David Park for his careful proofreading and professional English editing of this manuscript.

## Author Contributions

**Conceptualization:** Onuma Chaiwat.

**Data curation:** Chayanan Thanakiattiwibun.

**Formal analysis:** Onuma Chaiwat, Chayanan Thanakiattiwibun.

**Investigation:** Onuma Chaiwat, Mingkwan Wongyingsinn, Arunotai Siriussawakul, Pornpoj Pramyothin, Panita Limpawattana.

**Methodology:** Onuma Chaiwat, Arunotai Siriussawakul.

**Project administration:** Onuma Chaiwat, Arunotai Siriussawakul.

**Resources:** Weerasak Muangpaisan, Chalobol Chalermsri.

**Supervision:** Onuma Chaiwat.

**Validation:** Pornpoj Pramyothin, Panita Limpawattana.

**Writing – original draft:** Onuma Chaiwat, Mingkwan Wongyingsinn, Pornpoj Pramyothin, Poungkaew Thitisakulchai, Chayanan Thanakiattiwibun.

**Writing – review & editing:** Onuma Chaiwat, Weerasak Muangpaisan, Chalobol Chalermsri, Arunotai Siriussawakul, Pornpoj Pramyothin, Poungkaew Thitisakulchai, Panita Limpawattana, Chayanan Thanakiattiwibun.

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
