## [Decision Letter · Decision Letter 0]

15 Jun 2021

PONE-D-21-00876

The validity of a simple screening tool for sarcopenia in surgical patients

PLOS ONE

Dear Dr. Chaiwat,

Thank you for submitting your manuscript to PLOS ONE. After careful consideration, we feel that it has merit but does not fully meet PLOS ONE’s publication criteria as it currently stands. Therefore, we invite you to submit a revised version of the manuscript that addresses the points raised during the review process.

Both reviewers have raised a number of questions regarding the study design and analysis. Please ensure that you respond thoroughly to all of the reviewers' points when preparing your revised manuscript.

We look forward to receiving your revised manuscript.

Kind regards,

Jamie Males

Staff Editor

PLOS ONE

Journal Requirements:

2. Some text appears to be missing in your Ethics Statement: 'All patients or, if applicable, provided informed consent in writing'. We believe you may have omitted text regarding the legal guardian of the participants. Please amend this statement as necessary

Reviewers' comments:

Reviewer's Responses to Questions

**Comments to the Author**

1. Is the manuscript technically sound, and do the data support the conclusions?

Reviewer #1: Partly

Reviewer #2: Yes

Reviewer #3: Yes

2. Has the statistical analysis been performed appropriately and rigorously? 

Reviewer #1: No

Reviewer #2: Yes

Reviewer #3: N/A

3. Have the authors made all data underlying the findings in their manuscript fully available?

Reviewer #1: No

Reviewer #2: Yes

Reviewer #3: Yes

4. Is the manuscript presented in an intelligible fashion and written in standard English?

Reviewer #1: No

Reviewer #2: Yes

Reviewer #3: No

5. Review Comments to the Author

Reviewer #1: 1. abstract: line27: sentence is not understandable ; methods section should be re-written to be more accurate.

2. Key words : assessment tool should be precised

3. Main manuscript : the exclusion criteria "patients unable to walk or stand up" is a bias and the prevalence of sarcopenia is underestimated. Those patients are the most fragile and they were not included in the study.

4. "presence of pacemaker" : a pacemaker is not a contraindication for BIA, see the reference in pubmed https://pubmed.ncbi.nlm.nih.gov/29525512/

5. The exclusion criteria lead the author to not included a large number of patients, the most fragil especially : "taking medication, herb and/or hormone that affected muscle mass"

6. Patients with an alcohol consumption were not included but the variable was collected (line 112) without any explanations on the method

7. History of weigth loss was also collected but not displayed in tables

8. The research team chose to measure handgrip strength on the dominant arm, but what's happen in case of stroke for example. It's better to perform 3 measurements of each arm and then keep the higher one.

9. line 156 "four combination formulas" : not clear enougth and the explanation is in the result section instead of the method section.

10. The research team should explain why they began their algorithm with walk test instead of grip strength as the EWGSOP2 guidelines. Indeed, handgrip strength is an easy and well accepted test as the chair rise. The patients that have a positive screening for sarcopenia can be care safety as they are sarcopenic, in cancer context especially.

Reviewer #2: I read the text with great interest. Authors explored the operational definition of sarcopenia according to AWGS criteria in a vulnerable patient population. % 34 of elderly cancer patients had sarcopenia and the sarcopenia in these patient population was significantly related with poor outcomes. Authors introduced a simpler algorithm for predicting sarcopenia by excluding the muscle mass measurement and incorporating MNA-SF tool. Presence of cachexia and secondary sarcopenia in cancer patients emphasized the importance of nutritional screening.

It is a prospective, well organized study and authors may consider the following comments:

1. The authors named the simpler method of defining sarcopenia, as a validated tool. However I think it will be better to present this as a simle algorithm not a validated tool. Actually authors are defining a simpler form of algorithm by incorporating MNA-SF instead of muscle mass into the operational definition of sarcopenia. I recommend to change as a simple algorithm instead of a validated tool in the title and in the text.

2. AWGS has updated consensus in 2019 and changed the cut-off values for handgrip strength for men (<28kg) and also changed cut-off criteria for low physical performance for 6-m walk (<1.0 m/s). In the text, the old AWGS cut off points reported in 2014 were used. I wonder why authors did not use the cut-offs defined in updated AWGS.

3. In table 1 surgical oncologic patients with sarcopenia had poorer outcomes such as lower Barthel index score at 3 months of discharge, higher mortality rate at 3 months and 1 year after discharge again. These outcome findings are valuable and have scientific impact. But these outcomes are not mentioned in result section and not discussed in discussion section. I recommend to highlight these outcomes both in result and discussion sections.

4. I also wonder if sarcopenia, when defined according to your new algorithm, is related with poor outcome measures?

5. Authors defining a flowchart for screening of sarcopenia in Figure 1. Figure 1 represents the sarcopenia screening algorithm of EWGSOP in 2010 by starting with gait speed only. Unlike EWGSOP, AWGS recommends measuring both muscle strength (handgrip strength) and physical performance (usual gait speed) as the screening test. Actually the final result does not change, but still the authors can rearrange figure 1 according to the AWGS algorithm.

Minor points

Abstract

1. Line 26: muscle mass and muscle functions (strength and function). Please use physical performance term instead of function in parentheses.

2. Line 27: Please change the word immobilization with immobile

Result section

1 Line 171-173: Please rewrite the sentence, it is not clear ‘’ Of those, only 84 subjects demonstrated low muscle mass below the recommended cutoff value for a prevalence of sarcopenia in this cohort of 34% .’’

3. Line 175-176: Presarcopenia was found in another 34 patients (40% ); however, these patients were included in the non–sarcopenia group for all other analyses’’. Here the term presarcopenia refers to patients only with low muscle mass not accompanying low muscle strength and muscle performance. I couldn't catch the rate 40% . 34 out of 104 patients have low muscle mass ?? In addition the term presarcopenia was included in EWGSOP in 2010, not addressed in revised EWGSOP, AWGS in 2014 and updated AWGS in 2019. You may remove this information from method and result sections.

4. Line 190: Please write open form of the abbreviation CCI

Discussion

1. Line:246-248: ‘’Previous study from China in oncology surgical patients defined sarcopenia by the combination of low muscle mass and/ or low muscle strength and low physical performance’’. This information lacks the reference. You may add reference or may remove this information, the discussion is long anyway.

2. Dİscussion is too long, from lines 286 to 315 should be shortened

Reviewer #3: Dear Author,

I read the article entitled “The validity of a simple screening tool for sarcopenia in surgical patients” with great interest. Screening and diagnosis of sarcopenia is important in geriatric assessment. Sarcopenia, the age-associated loss of skeletal muscle mass, has been postulated to be a major factor in the strength decline with aging. Moreover, sarcopenia is related to functional impairment, disability, falls, and loss of independence in older adults. In clinical use there are some screening tools for sarcopenia. Therefore, new tools may be tried for screening and / or diagnostic purposes. Accordingly, this study provides important data. A lot of efforts have been put together for this study. I congratulate the authors for the study because this is a worthwhile study and can be considered for publication. However, there are some major points that should be revised substantially. Please find my comments below.

Abstract

1- In the background part its written that ‘’...muscle functions (strength and function)": this phrase is confusing? What do you indicate by function within the parenthesis?

2- ‘’ ….which is costly and immobilization’’ This phrase also makes no sense.

3- In line 27, We differentiate between "assessment" and "screening". Assessment is used for diagnosis whereas screening is for screening. That is to further select cases for assessment (diagnostic) protocols. Please express your intend in a correct way.

4- In line 28, What kind of an assessment tool is this? Is it something you suggest or already has been suggested in previous studies? If this was your first-time suggestion, then you should give details on its components.

5- In line 29, ‘’ ...diagnosis performance..’’ It should be "diagnostic performance" In general there are flaws in use of English. English should be edited by a native speaker preferably

6- In line 29, Instead of elderly, please use "older adults".

7- In the results part line 36, malnutrition and underweight are very inter-related. I do not think that they can be included as independent variables within the same regression analysis. Please clarify.

8- In line 39, ‘’….risk of malnutrition & malnutrition and/ or abnormal physical performance.’’ As far as I understood, it is better to write as “...risk of malnutrition/malnutrition". Also what do you mean by writing “...and/or .."

9- In conclusion part, line 43: We do not screen sarcopenia by muscle mass measurement. The authors might have had a confusion in this regard.

10- In the keywords, assessment tool is not the right word here. Consider screening tool.

Background

1- Two right square brackets in some references, please correct them.

2- In line 58 ‘’ Recent guidelines from the American College of Surgeons emphasize the importance of assessing sarcopenia prior to oncologic surgery in elderly patients.’’ In the referenced guideline they were not suggest assessing sarcopenia prior the surgery. They recommended document functional status, history of falls and frailty.

3- In line 62-65 ‘’ In essence….’’ This sentence is difficult to understand. Please describe better.

4- In line 69 ‘’… muscle quantity and quality’’ instead of "and", you should use "and/or".

5- In line 71-75 ‘’ Concerning the assessment of muscle mass….’’ Here, BIA comes forward due to its practical and portable application without exposure to any radiological harms. The authors are recommended to emphasize this fact, they may consider PMID: 28414253 to refer for this information.

6- In line 80 its written ‘assessment tool’. not assessment. Please be careful in your statements, particularly if your aim is to suggest a "screening" tool. Please review and correct your manuscript in terms of incorrect use of "assessment" word instead of screening.

7- In line 83-86 ‘’ Malnourished surgical……’’ After this sentence, the authors should develop the underlying the logic why they intended to use malnutrition as a component of sarcopenia screening in these patients.

8- Its written that the aims of this study were to design and validate the diagnosis performance of a simple assessment tool for screening sarcopenia in elderly cancer patients. There are some simple screening tools, such as SARC-F (PMID: 27066316 ). The authors should denote particularly SARC-F, as it is a very simple and convenient tool that demonstrated ability to predict adverse outcomes and sarcopenia, esp. the probable sarcopenia (PMID: 27066316 , PMID: 30272090). It has also potential other applications besides sarcopenia, such as frailty (PMID: 33786561). Moreover, SARC-F has been reported to screen for sarcopenia by application of alternative/lower cut-off scores (https://doi.org/10.1007/s12603-021-1617-3). SARC-F is also suggested by EWGSOP2 for "formal screening" and a project to widen its use has been endorsed by EuGMS (https://doi.org/10.1007/s41999-017-0003-5). Also, consider use of SARC-F by AWGS recommendation. These information on SARC-F should be noted in few sentences in order to avoid from a biased presentation of the literature. Moreover, I would suggest including at least a sentence on Ishii screening tool (PMID: 24450566). Also, SARc-CAlF which incorporates calf circumference as an indirect measure of muscle mass should be noted with few sentences (PMID: 27650212 and PMID: 30379299).

Materials and Methods

1- Its written that it is longitudinal and cross–sectional study. How can a study be both "longitudinal" and cross-sectional? It is conflicting.

2- In line 102 ‘Patients unable to walk or stand up were excluded.’ This makes that the proposed screening tool cannot be applied to those that are unable to walk, etc. This should be signified in the limitations section of the Discussion. Also, the authors should describe why they excluded these subjects.

3- In line 106 about the BIA measurement, Edema/ major fluid electrolyte abnormalities also precludes BIA assessment. Have you excluded those as well?

4- In line 118, the scoring categorization of MNA-SF should be integrated.

5- In line 119-120 measuring muscle mass with BIA, Is this appendicular muscle mass or total? You should specify.

6- In line 133-137 about the definition of sarcopenia, these are somewhat older references to define/diagnose sarcopenia. As authors would know, there are more updates diagnostic recommendations on sarcopenia both in Europe and Asia. This should be noted and discussed in the discussion section as a limitation of the study. Also, I guess the reference 27 should be corrected as reference 6.

7- In statistical analysis, have you checked normality? And how?

Results

1- In line 177 and 179 ‘Eleven patients..., five patients...’ Give % to allow readers understanding the impact of sarcopenia in decision processes better.

2- In line 190 ‘ …the sarcopenic group had a significantly lower number of patients with DM and DLP.’ This should be discussed with few sentences. Hypothetically sarcopenia should be related with higher DM and dyslipidemia due to loss of metabolic active muscle tissue. However, authors should clarify that sarcopenia is reported associated with low DM if muscle mass is adjusted by height2 but with higher DM if muscle is adjusted by weight/BMI (may refer https://doi.org/10.1016/j.eurger.2015.12.012).

3- In multivariate analysis, the authors should specify the dependent variable and independent variables in the regression analysis. This should also be given in the Table with a footnote.

4- In line 196, Barthel index is non-significant. Needless to state here.

5- About the formula 1, The authors should acknowledge that there is "probable sarcopenia" diagnosis that can be already made by solely measuring hand grip strength. The authors should specify that they are indicating confirmed sarcopenia or sarcopenia definitions that incorporate muscle mass.

6- Authors are recommended to give AUC values for the formulas as well.

7- In line 216, its written that EWGSOP2 demonstrated the highest specificity. Do you indicate confirmed sarcopenia or probable sarcopenia definitions of EWGSOP2 here?

Discussion

1- Discussion is too long. The authors should first outline main findings of their study and then discuss the findings by comparing with the similar studies in the literature.

2- In line 236 ‘In addition to the prevalence of sarcopenia being 34% , the prevalence rate increased with age...’ : I do not think that such detailed discussion on prevalence of sarcopenia in this study and comparison with others is needed. The objective of this study is NOT to report prevalence of sarcopenia. Your objective is to evaluate screening ability of the formulas you suggested. Therefore, shorten the Discussion and be sure that you discuss your findings around your main objective.

3- In line 263 ‘Regarding the factors related to sarcopenia, we found older age, malnutrition, and underweight status to be significantly associated with sarcopenia.’ : Refer to my previous comment on sarcopenia prevalence of sarcopenia/discussion. Concentrate on your main findings, not the secondary outputs. Just few sentences instead of this huge paragraph would be enough.

4- In line 295 they mentioned about SARC-F. My recommendations to note on SARC-F and Ishii tool in the Introduction sections may be detailed and answered at this part of the Discussion section. Nevertheless, some introductory sentences on SARC-f and Ishii should be stated in the Introduction section as well.

6. PLOS authors have the option to publish the peer review history of their article (what does this mean?). If published, this will include your full peer review and any attached files.

Reviewer #1: No

Reviewer #2: **Yes: **Firuzan Fırat Özer

Reviewer #3: No

---

## [Author Response · Author response to Decision Letter 0]

9 Jul 2021

Journal Requirements:

Response: We corrected as recommended.

2. Some text appears to be missing in your Ethics Statement: 'All patients or, if applicable, provided informed consent in writing'. We believe you may have omitted text regarding the legal guardian of the participants. Please amend this statement as necessary.

Response: We added these phrases.

“All procedures performed in studies involving human participants were in accordance with the 1964 Helsinki declaration and its later amendments or comparable ethical standards. This study was approved by the Institutional Review Board of the faculty of Medicine Siriraj Hospital (Si 101/2017). All patients or their legal guardians provided informed consent in writing. The analysis used anonymous clinical data that were obtained after each patient agreed to intervention by written consent.”

Response: We corrected as recommended.

“This research project was supported by Faculty of Medicine Siriraj Hospital, Mahidol University, Grant Number (IO) R016034004. The funders had no role in study design, data collection, and analysis, decision to publish, or preparation of the manuscript.”

 

Reviewers' comments:

Reviewer's Responses to Questions

Comments to the Author

1. Is the manuscript technically sound, and do the data support the conclusions?

Reviewer #1: Partly

Reviewer #2: Yes

Reviewer #3: Yes

2. Has the statistical analysis been performed appropriately and rigorously?

Reviewer #1: No

Reviewer #2: Yes

Reviewer #3: N/A

3. Have the authors made all data underlying the findings in their manuscript fully available?

Reviewer #1: No

Reviewer #2: Yes

Reviewer #3: Yes

4. Is the manuscript presented in an intelligible fashion and written in standard English?

Reviewer #1: No

Reviewer #2: Yes

Reviewer #3: No

5. Review Comments to the Author

Reviewer #1: 

1. abstract: line27: sentence is not understandable; methods section should be re-written to be more accurate.

Response: Abstract: We add some words to explain this sentence. (Page 2, Line 34-35)

Methods: We corrected the information in methods to be more accurate and clearer. We deleted this sentence “The validity of four combination formulas used for a diagnosis of sarcopenia compared to AWGS definition was presented” to make the method more understandable. 

2. Key words: assessment tool should be precised.

Response: We removed the word “assessment tool” and added “screening tool” as recommended by the reviewer#3. (Page 2, Line 47)

3. Main manuscript: the exclusion criteria "patients unable to walk or stand up" is a bias and the prevalence of sarcopenia is underestimated. Those patients are the most fragile and they were not included in the study.

Response: We realized about this limitation and addressed this in the limitation. Due to the limitation of funding and resources, we used the BIA from this company (Tanita MC–780U Multi Frequency Segmental Body Composition Analyzer; Tanita Corporation, Tokyo, Japan), this BIA can measure muscle mass only in the standing posture. We excluded 278 patients, 156 patients refused to participate and 122 (44%) patients cannot walk or stand up.

4. "presence of pacemaker”: a pacemaker is not a contraindication for BIA, see the reference in pubmed https://pubmed.ncbi.nlm.nih.gov/29525512/

Response: Thank you for this recommendation, however, at that time that we commenced this study, pace makers were the contraindication according to the manufacturer’s manual (https://www.tanita.com, page 4). Moreover, no patients were excluded from the presence of pacemaker.

5. The exclusion criteria lead the author to not included a large number of patients, the most fragil especially: "taking medication, herb and/or hormone that affected muscle mass"

Response: Yes, we agreed to this comment.

Nevertheless, in this study no patients were excluded regarding "taking medication, herb and/or hormone that affected muscle mass”.

6. Patients with an alcohol consumption were not included but the variable was collected (line 112) without any explanations on the method

Response: We excluded only patients who had consumed alcohol and/or had exercised strenuously within 12 hours prior to BIA measurement. We did collect the data regarding the history of alcohol consumption.

7. History of weight loss was also collected but not displayed in tables

Response: We collected the history of weight loss to define malnutrition by Mini Nutritional Assessment–Short Form (MNA®–SF) and reported as “malnutrition” in Table 1.

8. The research team chose to measure handgrip strength on the dominant arm, but what's happen in case of stroke for example. It's better to perform 3 measurements of each arm and then keep the higher one.

Response: Thank you for the comment and we agreed but we could not change at this time, for the next project we will consider this point. There were 11 stroke patients, we measured the handgrip in the non-weakness arm. 

9. line 156 "four combination formulas": not clear enough and the explanation is in the result section instead of the method section.

Response: We totally agreed. We moved the whole section from the results to the method section. (Page 7, Line 154-164)

10. The research team should explain why they began their algorithm with walk test instead of grip strength as the EWGSOP2 guidelines. Indeed, handgrip strength is an easy and well accepted test as the chair rise. The patients that have a positive screening for sarcopenia can be care safety as they are sarcopenic, in cancer context especially.

Response: We initially designed the protocol to assess sarcopenia according to the EWGSOP recommendation (ref 5.) at that time before the publication of EWGSOP2 guideline and it was recommended to start with gait speed.

Reference:

5. Cruz-Jentoft AJ, Baeyens JP, Bauer JM, Boirie Y, Cederholm T, Landi F, et al. Sarcopenia: European consensus on definition and diagnosis: Report of the European Working Group on Sarcopenia in Older People. Age Ageing. 2010; 39:412-23.

 

Reviewer #2: 

I read the text with great interest. Authors explored the operational definition of sarcopenia according to AWGS criteria in a vulnerable patient population. % 34 of elderly cancer patients had sarcopenia and the sarcopenia in these patient population was significantly related with poor outcomes. Authors introduced a simpler algorithm for predicting sarcopenia by excluding the muscle mass measurement and incorporating MNA-SF tool. Presence of cachexia and secondary sarcopenia in cancer patients emphasized the importance of nutritional screening.

It is a prospective, well organized study and authors may consider the following comments:

1. The authors named the simpler method of defining sarcopenia, as a validated tool. However I think it will be better to present this as a simple algorithm not a validated tool. Actually authors are defining a simpler form of algorithm by incorporating MNA-SF instead of muscle mass into the operational definition of sarcopenia. I recommend to change as a simple algorithm instead of a validated tool in the title and in the text.

Response: We changed the title to “The simpler screening tool for sarcopenia in surgical patients”. However, we designed the new screening tool and validated the diagnostic performance of this screening tool against the gold standard. So, we still used the verb “validate” in the text.

2. AWGS has updated consensus in 2019 and changed the cut-off values for handgrip strength for men (<28kg) and also changed cut-off criteria for low physical performance for 6-m walk (<1.0 m/s). In the text, the old AWGS cut off points reported in 2014 were used. I wonder why authors did not use the cut-offs defined in updated AWGS.

Response: We agreed. However, we commenced this study before the updated consensus in 2019. Moreover, we have been being in the submission process when AWGS 2019 was published. We will address this point in the limitation section. We did analysis for the new cut-off, the prevalence of sarcopenia was 41.8 % as compared to 33.5% with AWGS 2014. We also added the table 4b regarding the validity of combinations used to diagnose sarcopenia by using AWGS-2019 as a gold standard. The combination of low muscle strength and risk of malnutrition & malnutrition and/or abnormal physical performance (C3) still showed the highest sensitivity and accuracy (Table 4b).

3. In table 1 surgical oncologic patients with sarcopenia had poorer outcomes such as lower Barthel index score at 3 months of discharge, higher mortality rate at 3 months and 1 year after discharge again. These outcome findings are valuable and have scientific impact. But these outcomes are not mentioned in result section and not discussed in discussion section. I recommend to highlight these outcomes both in result and discussion sections.

Response: We have added information regarding the outcomes as recommended in methods (Page 7, Line 150-152), results (Page 14-15, Line 287-293) and discussion (Page 18-19, Line 366-376).  

4. I also wonder if sarcopenia, when defined according to your new algorithm, is related with poor outcome measures?

Response: Thank you, it was interesting and might be an opportunity to explore for the new project.

5. Authors defining a flowchart for screening of sarcopenia in Figure 1. Figure 1 represents the sarcopenia screening algorithm of EWGSOP in 2010 by starting with gait speed only. Unlike EWGSOP, AWGS recommends measuring both muscle strength (handgrip strength) and physical performance (usual gait speed) as the screening test. Actually the final result does not change, but still the authors can rearrange figure 1 according to the AWGS algorithm.

Response: We rearranged figure 1 as recommended.

Minor points

Abstract

1. Line 26: muscle mass and muscle functions (strength and function). Please use physical performance term instead of function in parentheses.

Response: We changed as recommended. (Page 2, Line 25)

2. Line 27: Please change the word immobilization with immobile.

Response: We changed as recommended (Page 2, Line 27)

Result section

1 Line 171-173: Please rewrite the sentence, it is not clear ‘’ Of those, only 84 subjects demonstrated low muscle mass below the recommended cutoff value for a prevalence of sarcopenia in this cohort of 34%.’’

Response: We rewrote to “Of those, only 84 subjects demonstrated low muscle mass below the recommended cutoff value resulted in a prevalence of sarcopenia in this cohort of 34%”.

3. Line 175-176: Presarcopenia was found in another 34 patients (40%); however, these patients were included in the non–sarcopenia group for all other analyses’’. Here the term presarcopenia refers to patients only with low muscle mass not accompanying low muscle strength and muscle performance. I couldn't catch the rate 40%. 34 out of 104 patients have low muscle mass ?? In addition the term presarcopenia was included in EWGSOP in 2010, not addressed in revised EWGSOP, AWGS in 2014 and updated AWGS in 2019. You may remove this information from method and result sections.

Response: We removed this information regarding “presarcopenia” as recommended. 

4. Line 190: Please write open form of the abbreviation CCI

Response: We added open form of CCI as recommended. (Page 11, Line 231)

Discussion

1. Line:246-248: ‘’Previous study from China in oncology surgical patients defined sarcopenia by the combination of low muscle mass and/ or low muscle strength and low physical performance’’. This information lacks the reference. You may add reference or may remove this information, the discussion is long anyway.

Response: We removed this information as recommended. 

2. Discussion is too long, from lines 286 to 315 should be shortened

Response: We shortened this information as recommended.

 

Reviewer #3: 

Dear Author,

I read the article entitled “The validity of a simple screening tool for sarcopenia in surgical patients” with great interest. Screening and diagnosis of sarcopenia is important in geriatric assessment. Sarcopenia, the age-associated loss of skeletal muscle mass, has been postulated to be a major factor in the strength decline with aging. Moreover, sarcopenia is related to functional impairment, disability, falls, and loss of independence in older adults. In clinical use there are some screening tools for sarcopenia. Therefore, new tools may be tried for screening and / or diagnostic purposes. Accordingly, this study provides important data. A lot of efforts have been put together for this study. I congratulate the authors for the study because this is a worthwhile study and can be considered for publication. 

However, there are some major points that should be revised substantially. Please find my comments below.

Abstract

1- In the background part its written that ‘’...muscle functions (strength and function)": this phrase is confusing? What do you indicate by function within the parenthesis?

Response: We changed to “physical performance”. (Page 2, Line 25)

2- ‘’ ….which is costly and immobilization’’ This phrase also makes no sense.

Response: We changed to “which is costly and immobile” as recommended by the 1st reviewer.

3- In line 27, We differentiate between "assessment" and "screening". Assessment is used for diagnosis whereas screening is for screening. That is to further select cases for assessment (diagnostic) protocols. Please express your intend in a correct way.

Response: Thank you for this correction. We changed the word “assessment” to “screening”. (Page 2, Line 27)

4- In line 28, What kind of an assessment tool is this? Is it something you suggest or already has been suggested in previous studies? If this was your first-time suggestion, then you should give details on its components.

Response: Yes, it was the first time suggestion for the screening tool but it was the aim to create this screening tool for screening sarcopenia without muscle mass measurement. We addressed its components in the results part of the abstract.

5- In line 29, ‘’ ...diagnosis performance..’’ It should be "diagnostic performance" In general there are flaws in use of English. English should be edited by a native speaker preferably

Response: Thank you for the correction. The manuscript was edited by the native speaker from our institution, however, there might be something wrong left. We will pay more attention to the gramma next time.

6- In line 29, Instead of elderly, please use "older adults".

Response: We changed as recommended. (Page 2, Line 30)

7- In the results part line 36, malnutrition and underweight are very inter-related. I do not think that they can be included as independent variables within the same regression analysis. Please clarify.

Response: It might not be very inter-related. Underweight patients might not have malnutrition especially in Thai population. Malnutrition according to MNA-SF, in addition to weight loss, relied on the history of eating, mobilization, stress and neuropsychological problems.

8- In line 39, ‘’….risk of malnutrition & malnutrition and/ or abnormal physical performance.’’ As far as I understood, it is better to write as “...risk of malnutrition/malnutrition". Also what do you mean by writing “...and/or .."

Response: and/or means 3 alternative ways (shown in Table 3, C3). 

1. Low muscle strength + Low physical performance + risk of malnutrition/malnutrition

2. Low muscle strength +risk of malnutrition/malnutrition

3. Low physical performance +risk of malnutrition/malnutrition

9- In conclusion part, line 43: We do not screen sarcopenia by muscle mass measurement. The authors might have had a confusion in this regard.

Response: We changed the sentence to “The screening of sarcopenia can be performed using a simpler screening tool”. (Page 2, Line 44)

10- In the keywords, assessment tool is not the right word here. Consider screening tool.

Response: We changed to “screening tool”. (Page 2, Line 47)

 

Background

1- Two right square brackets in some references, please correct them.

Response: We corrected as recommended.

2- In line 58 ‘’ Recent guidelines from the American College of Surgeons emphasize the importance of assessing sarcopenia prior to oncologic surgery in elderly patients.’’ In the referenced guideline they were not suggest assessing sarcopenia prior the surgery. They recommended document functional status, history of falls and frailty.

Response: We removed this sentence. 

3- In line 62-65 ‘’ In essence….’’ This sentence is difficult to understand. Please describe better.

Response: We rewrote to “In essence, each deﬁnition proposed to date deﬁnes sarcopenia as a state of decreased skeletal muscle mass and muscle function. Muscle function can be divided into those that require both muscle strength and physical functionality or only one of these elements”. (Page 3, Line 61-64)

4- In line 69 ‘’… muscle quantity and quality’’ instead of "and", you should use "and/or".

Response: We changed as recommended. (Page 3, Line 68)

5- In line 71-75 ‘’ Concerning the assessment of muscle mass….’’ Here, BIA comes forward due to its practical and portable application without exposure to any radiological harms. The authors are recommended to emphasize this fact, they may consider PMID: 28414253 to refer for this information.

Response: We added this sentence as recommended. (Page 4, Line 75-76)

Reference:

16. Yilmaz O, Bahat G. Suggestions for assessment of muscle mass in primary care setting. Aging Male. 2017;20:168-9.

6- In line 80 its written ‘assessment tool’. not assessment. Please be careful in your statements, particularly if your aim is to suggest a "screening" tool. Please review and correct your manuscript in terms of incorrect use of "assessment" word instead of screening.

Response: Thank you. We changed as recommended. (Page 4, Line 98)

7- In line 83-86 ‘’ Malnourished surgical……’’ After this sentence, the authors should develop the underlying the logic why they intended to use malnutrition as a component of sarcopenia screening in these patients.

Response: We moved this sentence “Since malnutrition and malignancy were factors that contribute to sarcopenia development, a simple screening tool for screening sarcopenia in patients who have cancer might be possible by incorporating malnutrition and underweight as screening factors.” (Page 4, Line 86-89)

8- Its written that the aims of this study were to design and validate the diagnosis performance of a simple assessment tool for screening sarcopenia in elderly cancer patients. There are some simple screening tools, such as SARC-F (PMID: 27066316 ). The authors should denote particularly SARC-F, as it is a very simple and convenient tool that demonstrated ability to predict adverse outcomes and sarcopenia, esp. the probable sarcopenia (PMID: 27066316 , PMID: 30272090). It has also potential other applications besides sarcopenia, such as frailty (PMID: 33786561). Moreover, SARC-F has been reported to screen for sarcopenia by application of alternative/lower cut-off scores (https://doi.org/10.1007/s12603-021-1617-3). SARC-F is also suggested by EWGSOP2 for "formal screening" and a project to widen its use has been endorsed by EuGMS (https://doi.org/10.1007/s41999-017-0003-5). Also, consider use of SARC-F by AWGS recommendation. These information on SARC-F should be noted in few sentences in order to avoid from a biased presentation of the literature. Moreover, I would suggest including at least a sentence on Ishii screening tool (PMID: 24450566). Also, SARc-CAlF which incorporates calf circumference as an indirect measure of muscle mass should be noted with few sentences (PMID: 27650212 and PMID: 30379299).

Response: We added as recommended. (Page 4, Line 90-96)

Materials and Methods

1- Its written that it is longitudinal and cross–sectional study. How can a study be both "longitudinal" and cross-sectional? It is conflicting.

Response: We agreed with the prospective longitudinal study. (Page 5, Line 103)

2- In line 102 ‘Patients unable to walk or stand up were excluded.’ This makes that the proposed screening tool cannot be applied to those that are unable to walk, etc. This should be signified in the limitations section of the Discussion. Also, the authors should describe why they excluded these subjects.

Response: We have addressed this in the limitation section of the discussion and add the sentence “because the bioimpedance analysis (BIA) (Tanita MC–780U Multi Frequency Segmental Body Composition Analyzer; Tanita Corporation, Tokyo, Japan) can measure muscle mass only in the standing position”. (Page 5, Line 113-114)

3- In line 106 about the BIA measurement, Edema/ major fluid electrolyte abnormalities also precludes BIA assessment. Have you excluded those as well?

Response: We performed the BIA measurement in the outpatients who visited the preoperative clinic so we did not exclude the patients who had edema/ major fluid electrolyte abnormalities.

4- In line 118, the scoring categorization of MNA-SF should be integrated.

Response: We added as recommended. (Page 6, Line 131-132)

5- In line 119-120 measuring muscle mass with BIA, Is this appendicular muscle mass or total? You should specify.

Response: We added “appendicular skeletal muscle mass” at that sentence. (Page 6, Line 133)

6- In line 133-137 about the definition of sarcopenia, these are somewhat older references to define/diagnose sarcopenia. As authors would know, there are more updates diagnostic recommendations on sarcopenia both in Europe and Asia. This should be noted and discussed in the discussion section as a limitation of the study. Also, I guess the reference 27 should be corrected as reference 6.

Response: We corrected the reference and added the information in the limitation section

7- In statistical analysis, have you checked normality? And how?

Response: We added this information in the statistical analysis section.

“The normally distribution was tested by histogram and the Kolmogorov-Smirnov test at P > 0.05.”

Results

1- In line 177 and 179 ‘Eleven patients..., five patients...’ Give % to allow readers understanding the impact of sarcopenia in decision processes better.

Response: We added as recommended. (Page 9, Line 207-211)

2- In line 190 ‘ …the sarcopenic group had a significantly lower number of patients with DM and DLP.’ This should be discussed with few sentences. Hypothetically sarcopenia should be related with higher DM and dyslipidemia due to loss of metabolic active muscle tissue. However, authors should clarify that sarcopenia is reported associated with low DM if muscle mass is adjusted by height2 but with higher DM if muscle is adjusted by weight/BMI (may refer https://doi.org/10.1016/j.eurger.2015.12.012).

Response: Thank you for the suggestion. We added some discussion as recommended. (Page 16-17, Line 320-325)

3- In multivariate analysis, the authors should specify the dependent variable and independent variables in the regression analysis. This should also be given in the Table with a footnote.

Response: We have added a table of multivariate regression and footnote regarding adjusted variables. (Table 2)

4- In line 196, Barthel index is non-significant. Needless to state here.

Response: We removed this sentence as recommended.

5- About the formula 1, The authors should acknowledge that there is "probable sarcopenia" diagnosis that can be already made by solely measuring hand grip strength. The authors should specify that they are indicating confirmed sarcopenia or sarcopenia definitions that incorporate muscle mass.

Response: We added texts regarding the "probable sarcopenia” in the results section. (Page 13, Line 256-260)

6- Authors are recommended to give AUC values for the formulas as well.

Response: We added AUROC in the table 4 and results section.

7- In line 216, its written that EWGSOP2 demonstrated the highest specificity. Do you indicate confirmed sarcopenia or probable sarcopenia definitions of EWGSOP2 here?

Response: We did the analyses and reported as texts in the results section. (Page 13, Line 256-260)

Discussion

1- Discussion is too long. The authors should first outline main findings of their study and then discuss the findings by comparing with the similar studies in the literature.

Response: We did remove some irrelevant parts of the discussion as recommended.

2- In line 236 ‘In addition to the prevalence of sarcopenia being 34%, the prevalence rate increased with age...’ : I do not think that such detailed discussion on prevalence of sarcopenia in this study and comparison with others is needed. The objective of this study is NOT to report prevalence of sarcopenia. Your objective is to evaluate screening ability of the formulas you suggested. Therefore, shorten the Discussion and be sure that you discuss your findings around your main objective.

Response: We agreed and cut detail about the discussion regarding the prevalence

3- In line 263 ‘Regarding the factors related to sarcopenia, we found older age, malnutrition, and underweight status to be significantly associated with sarcopenia.’ : Refer to my previous comment on sarcopenia prevalence of sarcopenia/discussion. Concentrate on your main findings, not the secondary outputs. Just few sentences instead of this huge paragraph would be enough.

Response: We agreed and removed some information of this part as well.

4- In line 295 they mentioned about SARC-F. My recommendations to note on SARC-F and Ishii tool in the Introduction sections may be detailed and answered at this part of the Discussion section. Nevertheless, some introductory sentences on SARC-f and Ishii should be stated in the Introduction section as well.

Response: We did as recommend.

---

## [Decision Letter · Decision Letter 1]

11 Aug 2021

PONE-D-21-00876R1

A simpler screening tool for sarcopenia in surgical patients

PLOS ONE

Dear Dr. Chaiwat,

Thank you for submitting your manuscript to PLOS ONE. After careful consideration, we feel that it has merit but does not fully meet PLOS ONE’s publication criteria as it currently stands. Therefore, we invite you to submit a revised version of the manuscript that addresses the points raised during the review process.

Please consider the reviewers and editor comments.

We look forward to receiving your revised manuscript.

Kind regards,

Joao Felipe Mota

Academic Editor

PLOS ONE

Journal Requirements:

Additional Editor Comments (if provided):

The study has many limitations which are addressed in this new version. I also consider as an important concern the inclusion of patients with oedema, so the authors should mention it in the limitation section.

Reviewers' comments:

Reviewer's Responses to Questions

**Comments to the Author**

1. If the authors have adequately addressed your comments raised in a previous round of review and you feel that this manuscript is now acceptable for publication, you may indicate that here to bypass the “Comments to the Author” section, enter your conflict of interest statement in the “Confidential to Editor” section, and submit your "Accept" recommendation.

Reviewer #2: All comments have been addressed

Reviewer #3: (No Response)

2. Is the manuscript technically sound, and do the data support the conclusions?

Reviewer #2: Yes

Reviewer #3: Yes

3. Has the statistical analysis been performed appropriately and rigorously? 

Reviewer #2: Yes

Reviewer #3: Yes

4. Have the authors made all data underlying the findings in their manuscript fully available?

Reviewer #2: Yes

Reviewer #3: Yes

5. Is the manuscript presented in an intelligible fashion and written in standard English?

Reviewer #2: Yes

Reviewer #3: No

6. Review Comments to the Author

Reviewer #2: Most of my concerns are addressed, but the text still needs a minör revision:

1.Please prefer the term “older adults” instead of “elderly” throughout the text

2.The text still needs a rigorous editing,

3.My suggestions for some phrases that cause misunderstanding are as follows:

Introduction

Line 77-78: The sentence’’ the DXA and BIA have some limitation in terms of the accessibility, and costly equipment’’. Please add the plural suffix ‘’s’’ to the word limitation. Instead of ‘’costly equipman’’ please write ‘’cost’’ only.

Line 95: In the sentence …….any one tool……, please remove the word ‘’one’’, write as ….any tool…

Line 97: Instead of ‘’diagnosis performance’’, please write ‘’diagnostic performance’’

Material and methods

Line 128: Other collected data included……Here the term other refers to components of MNA-SF tool. But it is perceived as a different data apart from MNA-SF tool. I recommend combine with previous sentence and you may write as ‘’ Preoperative nutritional screening was performed using the Mini Nutritional Assessment - Short Form (MNA® 128 –SF), which included reduction in dietary intake within the past three months, body mass index (BMI), history of weight loss within the last three months, mobility, psychological stress and/ or acute disease within the past 3 months, and neuropsychological problems.’’

Results

Line 249: Please rewrite subheading, you may write as ‘’ Diagnostic performance of a simple tool for screening sarcopenia’’

Line 250-252: Please rewrite the sentence ‘’The combination of low muscle strength and risk of malnutrition & malnutrition and/ or abnormal physical performance (C3) showed the highest sensitivity and accuracy either the AWGS or the updated AWGS as the gold standards’’you may write as ‘’ The combination of low muscle strength and risk of malnutrition & malnutrition and/ or abnormal physical performance (C3) showed the highest sensitivity and accuracy when using the AWGS or the updated AWGS as the gold standards’’

Line 252-254:Please rewrite the sentence ‘’ The sensitivity, specificity, accuracy and AUROC were 81.0%, 78.4%, 79.3% and 0.8, respectively as compared to C1, C2 and C4 when using AWGS as a gold standard’’. In this sentence percent results are belong to C3, but C3 is not mentioned in the sentence, in addition sentence gives the statistical information only related to C3, there is not a comparision between the statistical results of the formulas. You may rewrite the sentence as ‘’ The sensitivity, specificity, accuracy and AUROC of C3 were 81.0%, 78.4%, 79.3% and 0.8, respectively when using AWGS as gold standard.’’

Line 264-266: with similar reasons as above mentioned you may rewrite the sentence as ‘’ The sensitivity, specificity, accuracy, and AUROC of C3 were 80% , 68. 5% , 73. 3% and 0.74 respectively when using the updated AWGS criteria as gold standard (Table 4b).’’

Discussion

Line 306-308: Please rewrite the sentence, you may rewrite as ‘’ The combination of low muscle strength and risk of malnutrition & malnutrition and/or abnormal physical performance (C3), demonstrated high sensitivity, specificity, and predictive power when validated against a consensus of the Asian Working Group for Sarcopenia (AWGS).

Line 329: Please write immobile instead of immobilization

Reviewer #3: *English should be improved in the whole writing.

Abstract

*Page 2, Line 27: ‘Non-portable’ should be used instead of immobile device.

*Page 2, Line 28: The aim of this study was to design and validate the diagnostic performance of a simple screening tool for screening sarcopenia

These formula are not suggested for diagnosis of sarcopenia. Therefore, ‘diagnostic’ term should not be used.

*Page 2, Line 32: The details about this screening tool should be explained in methods part. Which parameters were considered for analyses, the details about formula should be briefly mentioned.

*Page 2, Line 37: Malnutrition and underweight status are surely expected to be strongly related. Therefore, before putting into the same regression analysis, multicollinearity should be checked.

Background

*Page 3, Line 57: In whole writing, elderly term should be changed to older adults.

*Page 3, Line 62: Muscle function can be divided into those that require both muscle strength and physical functionality or only one of these elements.

Functionality should be revised as ‘performance’.

*Introduction should be shortened.

Materials and methods

*Page 5, Study population exclusion criteria: Edema can also affect the BIA results, therefore, measurements without regarding edema status should also be stated as a limitation of the study.

*Page 7, Line 155: At risk and malnutrition and underweight BMI were considered to be factors for diagnosing sarcopenia.

Underweight BMI should be changed as ‘Being underweight by BMI’. Again, language revision should be done seriously.

Results

*Page 12, Line 249: Diagnosis performance a simple tool for screening sarcopenia

Diagnostic performance term should not be used. These formula were suggested for sarcopenia screening, so 'screening' and 'diagnosis' terms should not be used interchangeably.

*Page 13, Line 256: EWGSOP2 demonstrated the highest specificity (100%).

Whether the high specificity proposed by EWGSOP2 is for probable sarcopenia or confirmed sarcopenia should be stated.

Discussion

*Page 18, Line 355: Although, there was a slightly low specificity and PPV in female, we introduced a practical algorithm for a diagnosis of sarcopenia in elderly cancerous surgical patients without using muscle mass measurement (Fig 2).

The algorhythm the authors proposed for sarcopenia diagnosis seems problematic. The authors should decide whether they propose this formula for screening (finding cases) or assessment of sarcopenia. In addition, muscle mass measurement was included for diagnosis in the algorhythm. But they claimed that the algorhythm enables to diagnose sarcopenia without muscle mass measurement. There seems a paradox.

Conclusion

*Conclusion should be shortened. The first two sentences are the findings previously mentioned, unnecessary to repeat at the end.

7. PLOS authors have the option to publish the peer review history of their article (what does this mean?). If published, this will include your full peer review and any attached files.

Reviewer #2: **Yes: **Firuzan Fırat Özer

Reviewer #3: **Yes: **Gulistan Bahat

---

## [Author Response · Author response to Decision Letter 1]

5 Sep 2021

Dear editors,

We attached file the responses to the reviewers to this resubmission. 

Please find our point-by-point responses.

Regards,

Onuma Chaiwat

---

## [Editor Report · Decision Letter 2]

8 Sep 2021

A simpler screening tool for sarcopenia in surgical patients

PONE-D-21-00876R2

Dear Dr. Chaiwat,

We’re pleased to inform you that your manuscript has been judged scientifically suitable for publication and will be formally accepted for publication once it meets all outstanding technical requirements.

Kind regards,

Joao Felipe Mota

Academic Editor

PLOS ONE

---

## [Editor Report · Acceptance letter]

14 Sep 2021

PONE-D-21-00876R2 

A simpler screening tool for sarcopenia in surgical patients 

Dear Dr. Chaiwat:

I'm pleased to inform you that your manuscript has been deemed suitable for publication in PLOS ONE. Congratulations! Your manuscript is now with our production department. 

Kind regards, 

on behalf of

Dr. Joao Felipe Mota 

Academic Editor

PLOS ONE